# Genomic Characterization of an Extensively Drug-Resistant Extra-Intestinal Pathogenic (ExPEC) *Escherichia coli* Clinical Isolate Co-Producing Two Carbapenemases and a 16S rRNA Methylase

**DOI:** 10.3390/antibiotics11111479

**Published:** 2022-10-26

**Authors:** Mustafa Sadek, Alaaeldin Mohamed Saad, Patrice Nordmann, Laurent Poirel

**Affiliations:** 1Medical and Molecular Microbiology, Faculty of Science and Medicine, University of Fribourg, 1700 Fribourg, Switzerland; 2Department of Food Hygiene and Control, Faculty of Veterinary Medicine, South Valley University, Qena 83522, Egypt; 3Department of Zoonoses, Faculty of Veterinary Medicine, Zagazig University, Zagazig 44511, Egypt; 4INSERM European Unit (IAME), University of Fribourg, 1700 Fribourg, Switzerland; 5Swiss National Reference Center for Emerging Antibiotic Resistance (NARA), University of Fribourg, 1700 Fribourg, Switzerland; 6Institute for Microbiology, Lausanne University Hospital and University of Lausanne, 1015 Lausanne, Switzerland

**Keywords:** *Escherichia coli*, carbapenemase, plasmid, NDM, KPC

## Abstract

An extensively drug-resistant *Escherichia coli* clinical isolate (N1606) belonging to Sequence Type 361 was recovered from the urine of a patient hospitalized in Switzerland. The strain showed resistance to virtually all β-lactams including the latest generation antibiotics cefiderocol and aztreonam–avibactam. Whole genome sequencing revealed that it possessed two carbapenemase-encoding genes, namely *bla*_NDM-5_ and *bla*_KPC-3_, and a series of additional β-lactamase genes, including *bla*_CTX-M-15_ and *bla*_SHV-11_ encoding extended-spectrum β-lactamases (ESBLs), *bla*_CMY-145_ encoding an AmpC-type cephalosporinase, and *bla*_OXA-1_ encoding a narrow-spectrum class D ß-lactamase. Most of these resistance genes were located on plasmids (IncFII-FIA, IncX3, IncIγ, IncFII). That strain exhibited also a four amino-acid insertion in its penicillin-binding protein 3 (PBP3) sequence, namely corresponding to YRIN. Complete genome analysis revealed that this *E. coli* isolate carried virulence factors (*sitA, gad, hra, terC*, *traT*, and *cia*) and many other non-β-lactam resistance determinants including *rmtB*, *tet(A), dfrA17* (two copies), *aadA1, aadA5* (two copies), *sul1* (two copies), *qacE* (two copies), *qepA*, *mdf(A)*, *catA1*, *erm(B)*, *mph(A)*, and *qnrS1*, being susceptible only to tigecycline, colistin and fosfomycin. In conclusion, we described here the phenotypic and genome characteristics of an extensively drug-resistant (XDR) *E. coli* ST361 being recognized as an emerging clone worldwide.

## 1. Introduction

*Escherichia coli* is considered as the most important pathogen for humans, receiving special attention in the microbiological world, since it causes a wide range of severe infections in humans and animals such as urinary tract infections and other community-acquired infections [1]. *E. coli* has a high capacity to accumulate resistance genes acquired mostly through horizontal gene transfer, leading to phenotypic resistance to antibiotics that are normally very effective against this bacterial species [2]. Difficult-to-treat infections caused by carbapenem-resistant and carbapenemase-producing *E. coli* has become an issue of utmost importance which is increasingly recognized in recent years, being associated with increased morbidity and mortality [3]. Five main carbapenemases are identified among enterobacterial species, including in *E. coli* clinical isolates, namely the Ambler class A KPC-type ß-lactamases, the class B ß-lactamases of the NDM, VIM, or IMP types, and the class D ß-lactamase OXA-48 and its derivatives [4]. During the last decade, clinical isolates co-producing multiple carbapenemases have been reported in different countries [5,6,7,8,9,10]. Considering that most NDM-type-producing Enterobacterales (including *E. coli*) are highly resistant both to most β-lactams and non-β-lactams antibiotics, the recently developed aztreonam/avibactam (ATM-AVI) combo offers an alternative therapeutic of choice [11]. However, the emergence of ATM-AVI resistance is being increasingly reported among NDM-producing *E. coli* clinical isolates [11,12]. On the other hand, cefiderocol (FDC), a novel siderophore cephalosporin, has been recently developed. FDC is one of the latest generations of commercialized antibiotics with excellent antibacterial activity against a large variety of Gram negatives including carbapenem-resistant Enterobacterales, using its so-called “Trojan horse” unique strategy [13,14]. In this study, we report an extensively drug-resistant *E. coli* clinical isolate carrying a large array of resistance genes leading to resistance to all β-lactams including FDC and most non-β-lactams.

## 2. Material and Methods 

In November 2020, *E. coli* clinical strain N1606 was recovered from the human urine sample from a patient hospitalized in Switzerland. The isolate was first identified as *E. coli* using EnteroPluri-test (Liofilchem SRL, Roseto degli Abruzzi, Italy) that later was confirmed by whole genome sequencing (WGS). Two carbapenemase-encoding genes (*bla*_NDM_ and *bla*_KPC_) were detected using GeneXpert^®^ system (Cepheid, Sunnyvale, CA, USA), which is a molecular diagnostic platform. Antimicrobial susceptibility testing was then performed by using the disk diffusion method on Mueller–Hinton agar plates for selected antibiotics. Minimum inhibitory concentrations (MICs) were also determined using Etest strips (bioMérieux, La Balme-les-Grottes, France) on Mueller–Hinton agar plates at 37 °C except for aztreonam/avibactam (ATM-AVI), for which the MICs were determined using the broth microdilution (BMD) in cation-adjusted Mueller–Hinton broth (Bio-Rad, Marnes-la-Coquette, France), and AVI was tested at a fixed concentration of 4 μg/mL. MICs of fosfomycin were determined using AD fosfomycin agar dilution test (Liofilchem, Italy) following the manufacturer’s instructions. For FDC, MIC values were determined with the reference BMD method using iron-depleted cation-adjusted Mueller–Hinton (ID-CAMH) broth prepared following the protocol described by Hackel et al. [15]. The results were interpreted according to the latest EUCAST breakpoints (https://www.eucast.org/fileadmin/src/media/PDFs/EUCAST_files/Breakpoint_tables/v_12.0_Breakpoint_Tables.pdf (accessed on 14 February 2022)) [16]. MICs of colistin were determined using broth microdilution in cation-adjusted Mueller–Hinton broth (Bio-Rad), and results were interpreted according to the EUCAST/CLSI joint guidelines (www.eucast.org, (accessed on 14 February 2022)). *E. coli* ATCC 25922 was used as control for all testing.

Carbapenemase production was determined by using the biochemical Carba NP, NitroSpeed-Carba NP tests [17,18], and by using immunochromatographic NG-Test Carba5 assay (NG Biotech, France). The Rapid Polymyxin NP test and the Rapid Fosfomycin NP test were performed as described [19,20]. PCR screening was performed for the presence of the major carbapenemase genes (*bla*_KPC_, *bla*_OXA-48_, *bla*_NDM_, *bla*_VIM_, and *bla*_IMP_) as previously described [21]. Sanger sequencing of amplified carbapenemase genes was performed by Microsynth AG (Microsynth AG, https://www.microsynth.com, (accessed on 22 March 2022)) to identify the exact alleles.

Mating-out assays using the filter-mating method were performed as described previously (6). The *E. coli* N1606 isolate was used as a donor, and the azide-resistant *E. coli* J53 strain was used as the recipient. Briefly, both donor and recipient strains were cultured separately in LB broth. After incubation, the donor and recipient strains were mixed at a ratio of 1:9 (donor/recipient) and centrifuged, the supernatant was removed, and the pellets were resuspended in 200 μL LB broth, which was plated on a conjugation filter on an LB agar plate. The plate was incubated for 5 h at 37 °C. Transconjugants were selected on LB agar supplemented with sodium azide (100 mg/L) and imipenem (4 mg/L) for *bla*_NDM-5_ carrying plasmid or cefoxitin 1 mg/L for *bla*_KPC-3_ carrying plasmid or gentamicin 50 mg/L and amikacin 50 mg/L for methylase. Successful transconjugants were confirmed by performing antimicrobial susceptibility testing and PCR targeting the genes of interest.

The entire genome of *E. coli* N1606 was sequenced using a combination of the MiSeq and MinION (Oxford Nanopore Technologies, Oxford, United Kingdom) platforms. Briefly, the total genomic DNA (gDNA) was extracted using a QIAamp DNA minikit and QIAcube (Qiagen) according to the manufacturer’s instructions. For the MiSeq sequencing, a DNA library was constructed using the Nextera sample preparation with 2 × 150 bp paired-end reads (Illumina, San Diego, CA, USA) according to the manufacturer’s instructions. MinION sequencing was performed using the MinION Mk1C (Oxford Nanopore Technologies, Oxford, UK), and sequencing libraries were prepared using a native barcoding kit (EXP-NBD104; Oxford Nanopore Technologies, UK) and 1D chemistry Ligation Sequencing Kit (SQK-LSK109; Oxford Nanopore Technologies) and performed on a R9.4.1 Flow Cell (FLO-MIN106; Oxford Nanopore Technologies). Assembly of both Illumina short reads and Nanopore long reads were performed using the CLC Genomic Workbench (version 20.0.4; CLC Bio, Aarhus, Denmark). The resulting assembled sequences were analyzed for antimicrobial resistance genes, multilocus sequence typing (MLST), Serotyping and *fimH* subtyping, plasmid incompatibility groups, and plasmid MLST using the Center for Genomic Epidemiology server (http://www.genomicepidemiology.org/, (accessed on 17 May 2022)). The complete genome sequence of *E. coli* N1606 strain and all plasmids had been deposited at GenBank under BioProject ID: PRJNA762038. The complete nucleotide sequences of chromosome (Chr1606) and plasmids (p1606A, p1606B, p1606C, p1606D, p1606E, p1606F, and p1606G) were deposited as GenBank accession numbers CP083701, CP083702, CP083703, CP083704, CP083705, CP083706, CP083707, and CP083708, respectively.

## 3. Results and Discussion 

*E. coli* N1606 was highly resistant to all β-lactams (amoxicillin, ticarcillin, piperacillin, mecillinam, piperacillin–tazobactam, amoxicillin–clavulanate, ticarcillin–clavulanate, ceftazidime, cefotaxime, cefepime, temocillin, cefoxitin, imipenem, ertapenem, meropenem, aztreonam, but also to non-β-lactams including fluoroquinolones (ciprofloxacin, norfloxacin), aminoglycosides (gentamicin, amikacin, kanamycin, tobramycin, netilmicin), trimethoprim–sulfamethoxazole, nitrofurantoin, and chloramphenicol (Table 1). The isolate was tested negative using the Rapid Polymyxin NP test and the Rapid Fosfomycin NP test, indicating the susceptibility to colistin and fosfomycin, respectively.

MIC determinations confirmed the susceptibility to colistin and fosfomycin as well as the susceptibility to tigecycline (Table 1). Furthermore, *E. coli* N1606 was also resistant to all the recently β-lactam/β-lactamase inhibitor combinations such as ceftazidime/avibactam, imipenem/relebactam, meropenem/vaborbactam, and ceftolozane/tazobactam. It is noteworthy that resistance was also observed with the most recent promising therapeutic options such as FDC or aztreonam/avibactam. Interpretation was also performed following the EUCAST breakpoints, resistance being >2 mg/L for FDC and >4 mg/L for ATM, that latter being also applied here for ATM-AVI.

Carbapenemase production was evidenced by using the biochemical Carba NP and NitroSpeed-Carba NP tests [17,18]. Using the immunochromatographic NG-Test Carba5 assay (NG Biotech, Guipry-Messac, France), the production of both NDM and KPC carbapenemases was detected. Using PCR and sequencing of the corresponding amplicons (Microsynth, Balgach, Switzerland), NDM-5 and KPC-3 were identified. 

Mating-out assays were performed for the *E. coli* isolate N1606 with the aim to transfer the *bla*_NDM-5_ and *bla*_KPC-3_ genes. *bla*_NDM-5_ was transferred by conjugation, but *bla*_KPC-3_ was not, and the self-transmissible plasmid carrying *bla*_NDM-5_ was designated p1606A. By testing the *bla*_NDM-5_ positive transconjugant for antibiotic susceptibility, co-resistance to aminoglycosides (kanamycin, tobramycin, gentamicin, and amikacin) was observed, as was resistance to sulfonamides and tetracycline. Resistance to aminoglycosides was correlated with the acquisition of the 16S RNA methylase-encoding gene *rmtB*. PCR-based replicon typing (PBRT) analysis revealed that the *bla*_NDM-5 gene_ was localized on an IncFII–FIA plasmid scaffold. 

WGS analysis revealed that *E. coli* N1606 belonged to Sequence Type ST361 and to the O9:H30 serotype. The lineage in which *E. coli* ST361 belongs is not recognized so far as internationally widespread with very few reports either in human or animals, making it difficult to fully discuss and compare results from different epidemiological studies [22,23,24,25]. Very recently, an *E. coli* ST361 harboring the *bla*_NDM-5_ gene has been described from different human and animal sources in several European countries including Switzerland [26,27,28]. 

In addition to the *bla*_NDM-5_ and *bla*_KPC-3_ carbapenemase genes, *E. coli* N1606 possessed a series of antibiotic resistance genes (Table 2), including chromosomally encoded genes such as *mdf(A)* (macrolide)*, qepA* (fluoroquinolones)*, aadA1* (aminoglycosides)*, catA1* (chloramphenicol) and, *bla*_OXA-1_ (narrow-spectrum class D ß-lactamase), and plasmid-borne genes such as *bla*_CTX-M-15_, *bla*_SHV-11_ (ESBLs), *bla*_CMY-145_ (AmpC ß-lactamase), *bla*_TEM-1B_ (narrow-spectrum penicillinase), *rmtB* (16S rRNA methylase conferring pan-resistance to aminoglycosides)*, tet(A)* (tetracycline)*, aadA5* (2 copies) (streptomycin)*, dfrA17* (two copies) (trimethoprim)*, sul1* (two copies) (sulfonamides)*, qacE* (two copies) (quaternary ammonium)*, erm(B), mph(A)* (macrolides), and *qnrS1* (fluoroquinolones) (Table 2).

It also carried plasmid-borne virulence genes (*sitA, gad, hra, terC, traT, traT,* and *cia*), being therefore classified as an extraintestinal pathogenic strain (ExPEC). The genome of that strain was actually composed of a single chromosome (5,036.231 bp) and seven plasmids including p1606A (120,731 bp), p1606B (53,292 bp), p1606C (58,698 bp), p1606D (146,558 bp), p1606E (89,634 bp), p1606F (107,032 bp), and p1606G (2634 bp) (Table 2). 

The *bla*_NDM-5_ was located onto a 120,731-bp F-type plasmid (IncFII-FIA), namely p1606A, belonging to pMLST (F36:A4:B-) (Table 2). Previous studies have reported the diversity of plasmid types harboring the *bla*_NDM-5_ gene, including the IncF, IncFII, IncN, and IncX3 incompatibility groups [27,29,30,31,32]. Plasmid p1606A encoded multidrug resistance including all β-lactams (except aztreonam), aminoglycosides, sulfonamides, and tetracycline, since it harbored the *bla*_NDM-5_, *rmtB*, *aadA5*, *dfrA17*, *tet(A), bla*_TEM-1B_, *sul1*, and *qacE* resistance genes. The sequence of plasmid p1606A analyzed by using the Basic Local Alignment Search Tool (BLAST) provided by NCBI was highly similar to that of other *bla*_NDM-5_-positive IncF-type plasmids, namely p_dm702b_NDM5 (99% query coverage and 99.99% sequence identity; GenBank accession number CP095643), p_dm708c_NDM5 (99% query coverage and 99.97% sequence identity; GenBank accession number CP095644), and p_dm566_NDM5 (99% query coverage and 99.97% sequence identity; GenBank accession number CP095630) recently identified from *E. coli* isolates recovered in Bangladesh [33]. The *bla*_KPC-3_-carrying plasmid, namely p1606B, was 53,292 bp in size and belonged to the IncX3 incompatibility group, with *bla*_KPC-3_ being located into transposon Tn*4401* (△IS*3000-tnpR-tnpA-*IS*Kpn7-bla*_KPC-3_*-tnpA-*IS*Kpn6-*△IS*3000*), similarly to the structure identified on the KPC-3-IncX3 plasmid pCfr-145 (KY659388) recently identified among *Citrobacter freundii* isolates in Italy [34,35,36].

WGS-based analysis of the PBP3 sequence of *E. coli* isolate N1606 identified an insertion of four amino acids (YRIN) after residue 333 compared to the PBP3 sequence of the wild-type E. coli MG1655 reference strain (Genbank NC_000913.3). It was previously shown that four amino-acid insertions (such as YRIN or YRIK) are associated with increased resistance to ATM-AVI [12]. It is noteworthy that isogenic mutants possessing YIRN-inserted PBP3 displayed a slight two-fold reduced susceptibility to cefiderocol (from 0.06 to 0.125 mg/L) [37], However, more studies are needed to investigate the real impact (if any) of such modifications in PBP3 protein, the primary target of FDC, on the antibacterial activity of this novel cephalosporin.

We investigated here several genes of specific interest considering they had been previously shown to be involved in reduced susceptibility to FDC, including the *fiu* and *cirA* iron–catecholate transporter encoding genes, and other iron transport-related genes (*exbB, exbD, tonB3, baeS/R*) [38,39]. The analyzed genes (*fiu, cirA exbB, exbD, BaeS/R*) showed a wild-type sequence. A single amino acid mutation (L133P) was identified in TonB3, a component of the TonB3-ExbB3/D3 complex, which is providing energy required for FDC transport and associated with iron acquisition [40]. Mutations in the *tonB3* gene might impair the energy acquisition for FDC transport and iron availability, leading to reduced susceptibility to FDC as previously reported for other siderophore-conjugated antibiotics such as KP-736, BMS-180680, E-0702, pirazmonam, and U-78,608 [41].

To investigate whether the resistance to FDC observed in *E. coli* N1606 could be related to the production of acquired class A or C β-lactamases, MICs of FDC were determined in combination with avibactam (AVI) as an inhibitor of Ambler class A (including extended-spectrum β-lactamases (ESBLs), class C, and some class D ß-lactamases, including carbapenemases (e.g., KPC and OXA-48). Hence, MICs of FDC dropped from 64 to 4 mg/L when combined with AVI at a fixed concentration of 4 mg/L. This suggested that the high-level resistance to FDC in the tested isolate was likely due to the accumulation of a wide variety of different β-lactamases including carbapenemases, namely KPC-3, the ESBLs CTX-M-15 and SHV-11, and the AmpC enzyme CMY-145, as well as NDM-5 that had been previously shown to affect bacterial susceptibility toward FDC [42,43,44,45]. 

## 4. Conclusions 

We described here the genome characteristics of an extensively drug-resistant (XDR) *Escherichia coli* ST361 isolate co-carrying the *bla*_KPC-3_, *bla*_NDM-5_ and other resistance genes located on multiple plasmids. It is noteworthy that ST361 NDM-5-producing *E. coli* are increasingly identified worldwide, being either found in humans, the environment, or animals [28,46,47,48]. The identification of such an XDR isolate may constitute a serious challenge for resistance control and the clinical treatment of related infections. The emergence of such an XDR phenotype in the *E. coli* strain due to the accumulation of many plasmid-associated MDR determinants can be explained by the bacterial potential to acquire additional resistance traits through mobile genetic elements such as plasmids by horizontal gene transfer not only among E. coli strains but also to other Enterobacterales. This case here further underlines that FOS-containing treatment may be an option for treating infections associated with such resistant strain [49]. 

## Figures and Tables

**Table 1 antibiotics-11-01479-t001:** MICs of selected antibiotics for *E. coli* N1606 determined by Etest strips and broth microdilution.

Strain	MICs (mg/L)															
**N1606**	**CTX**	**CZD**	**CZA**	**FEP**	**C/T**	**IMI**	**IMI/REL**	**ETP**	**MEM**	**MEM/** **VAB**	**ATM**	**ATM/** **AVI**	**FDC**	**CST**	FOS	TGC
	>32	>256	>256	>256	>256	>32	>32	>32	>32	>32	>256	16	64	≤0.125	0.5	0.38

Abbreviations. CTX, cefotaxime; CZD, ceftazidime; CZA, ceftazidime/avibactam; FEP, cefepime; C/T, ceftolozane/tazobactam; IMI, imipenem; IMI/REL, imipenem/relebactam; ETP, ertapenem; MEM, meropenem; MEM/VAB, meropenem/vaborbactam; ATM, aztreonam; ATM/AVI, aztreonam/avibactam; FDC, cefiderocol; CST, colistin; FOS, fosfomycin; TGC, tigecycline.

**Table 2 antibiotics-11-01479-t002:** Characteristics of chromosome and plasmids harbored by *bla*_NDM-5_- and *bla*_KPC-3_-co-harboring *E. coli* clinical isolate N1606.

Genetic Elements	Size (bp)	MLST ^a^	pMLSTs (ABC Formula)	Plasmid Group	Antibiotic Resistance Gene(s)	PBP3	Virulence Factors	GenBank Accession no.
Chr1606	5,036,231	ST-361	-	*-*	*mdf(A), qepA, aadA1, catA1, bla_OXA-1_*	YRIN, Q227H, E353K, I532L	*SitA, gad, hra, terC*	CP083701
p1606A	120,731	-	[F36:A4:B-]	IncFII-FIA	*bla* _NDM-5_ *, rmtB, aadA5, dfrA17, tet(A), bla* _TEM-1B_ *, sul1, qacE*	*-*	*traT*	CP083702
p1606B	53,292	-	Unknown	IncX3	*bla* _KPC-3_ *, bla* _SHV-11_	*-*		CP083703
p1606C	58,698	-	Unknown	IncIγ	*bla* _CMY-145_	*-*		CP083704
p1606D	146,558	-	[F2:A-:B-]	IncFII	*bla* _CTX-M-15_ *, sul1, dfrA17, aadA5, erm(B), mph(A), qnrS1, qacE*	*-*	*traT*	CP083705
p1606E	89,634	-		IncY	*-*	*-*		CP083706
p1606F	107,032	-	Unknown	IncI1-I(Alpha)	*-*	*-*	*cia*	CP083707
p1606G	2634	-		Col(BS512)	*-*	*-*		CP083708

^a^ MLST, -, not applicable.

## Data Availability

Not applicable.

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
