# Peer review of "Genomic Characterization of an Extensively Drug-Resistant Extra-Intestinal Pathogenic (ExPEC) Escherichia coli Clinical Isolate Co-Producing Two Carbapenemases and a 16S rRNA Methylase"

_antibiotics, 2022, doi:10.3390/antibiotics11111479_

Round 1
Reviewer 1 Report
The manuscript describes the characterisation of an E. coli isolate which produces multiple beta-lactamases, including two carbapenemases. The microbiological work is rigorous, but there are some gaps in the description of methods and the discussion does not include some interesting epidemiological issues.
Specific comments:
Introduction: The introduction can be improved by justifying the deep study of a multiple beta-lactamase-producing E. coli. The most interesting feature of the isolate is the cefiderocol resistance, which should be introduced.
Material and methods, from line 53: The isolate has been recovered from a hospitalised patient in a country with a low prevalence of carbapenemase-producing isolates of this species. It would be interesting to know if he has travelled abroad, how long he has been hospitalised and, most importantly, what antibiotic treatment he has received previously that could explain a possible selection.
Material and methods, lines83-89: The use of CEM tools for annotation and molecular typing could be summarised.
Results, line 110: according to Material and methods section, only ETest strips have been used for AST testing. How colistin has been testing? Since 2016, EUCAST doesn't recommend to use colistin gradient strips.
Results, line 117: as far as I know, BioMérieux has not a strip with aztreonam/avibactam. How this antibiotic has been tested?
Results, line 123: the description of the PCR for blaKPC and blaNDM genes has not been included in material and methods section.
Results, lines 125-l34: the information in the text is redundant with the Table 1 for some of the determinants.
Results, line 125: Clone assignment aims to trace or relate to some cluster or origin. There are several ST361 isolates from Switzerland in Enterobase to what could be compared or related, some of animal origin. On the other hand, NDM-5 had been found in E. coli from river water in Switzerland (Bleichenbacher S, et al. Environmental dissemination of carbapenemase-producing Enterobacteriaceae in rivers in Switzerland. Environ Pollut. 2020 265(Pt B):115081). These epidemiological issues should be included in the discussion.
Results, lines 148-152: How the homology search has been carried out for the plasmids?
Author Response
Reviewer 1
The manuscript describes the characterisation of an E. coli isolate which produces multiple beta-lactamases, including two carbapenemases. The microbiological work is rigorous, but there are some gaps in the description of methods and the discussion does not include some interesting epidemiological issues.
Specific comments:
Introduction: The introduction can be improved by justifying the deep study of a multiple beta-lactamase-producing E. coli. The most interesting feature of the isolate is the cefiderocol resistance, which should be introduced.
- Added in the text according to the reviewer’s comment.
Material and methods, from line 53: The isolate has been recovered from a hospitalised patient in a country with a low prevalence of carbapenemase-producing isolates of this species. It would be interesting to know if he has travelled abroad, how long he has been hospitalised and, most importantly, what antibiotic treatment he has received previously that could explain a possible selection.
- Unfortunately, those data aren’t available.
Material and methods, lines83-89: The use of CEM tools for annotation and molecular typing could be summarised.
- It has now been summarized as suggested by the reviewer.
Results, line 110: according to Material and methods section, only Etest strips have been used for AST testing. How colistin has been testing? Since 2016, EUCAST doesn't recommend to use colistin gradient strips.
- It was already clarified in the text “MICs of colistin were determined using broth microdilution in cation-adjusted Mueller-Hinton broth (Bio-Rad) and results were interpreted according to the EUCAST/CLSI joint guidelines (www.eucast.org). E. coliATCC 25922 was used as quality control for all testing”.
Results, line 117: as far as I know, BioMérieux has not a strip with aztreonam/avibactam. How this antibiotic has been tested?
- It is now clarified in the revised version of the manuscript. “Minimum inhibitory concentrations (MICs) were also determined using Etest strips (bioMérieux, La Balme-les-Grottes, France) on Mueller-Hinton agar plates at 37°C except for aztreonam/avibactam (ATM-AVI) for which the MICs were determined using the broth microdilution (BMD) in cation-adjusted Mueller-Hinton broth (Bio-Rad, Marnes-la-Coquette, France) and AVI was tested at a fixed concentration of 4 μg/mL”.
Results, line 123: the description of the PCR for blaKPC and blaNDM genes has not been included in material and methods section.
- It is now included.
Results, lines 125-l34: the information in the text is redundant with the Table 1 for some of the determinants.
- This part has now been shortened.
Results, line 125: Clone assignment aims to trace or relate to some cluster or origin. There are several ST361 isolates from Switzerland in Enterobase to what could be compared or related, some of animal origin. On the other hand, NDM-5 had been found in E. coli from river water in Switzerland (Bleichenbacher S, et al. Environmental dissemination of carbapenemase-producing Enterobacteriaceae in rivers in Switzerland. Environ Pollut. 2020 265(Pt B):115081). These epidemiological issues should be included in the discussion.
- This has now been added as suggested by the reviewer.
Results, lines 148-152: How the homology search has been carried out for the plasmids?
- Using the Basic Local Alignment Search Tool (BLAST) provided by NCBI using a 99% identity cutoff. This information has been now added in the revised version.
Reviewer 2 Report
Sadek et al. isolated E. coli strain N1606, and showed its extensive resistance to a variety of antibiotics. They further sequenced N1606 using a combination of MiSeq and Nanopore reads. The manuscript is well-written and in its good shape.
Major comments:
1. This work sequenced the genome of an E. coli strain. Please briefly describe the library preparation process.
2. I would suggest using Galaxy (https://usegalaxy.org/) to assemble E. coli genomes. Galaxy offers free access. The analysis steps can be easily shared, and the analysis process is reproducible, the latter of which, in my opinion, matters significantly in this instance. The Galaxy communities published a preprint paper (https://doi.org/10.1101/347625) where an E. coli strain was fully sequenced and analyzed upon MiSeq and Nanopore reads.
3. I would suggest adding a table showing the antibiotics tested and the MIC numbers.
4. Please further discuss why N1606 has accumulated so many antibiotics-resistant genes.
Minor comments:
1. Page 2 line 54, "from human urine sample" → "from the human urine sample"
2. Page 2 line 71, "quality control" → "control".
3. Page 2 line 82, "2634 bp to 120731 bp" → "2,634 bp to 120,731 bp".
Author Response
Reviewer 2
Sadek et al. isolated E. coli strain N1606, and showed its extensive resistance to a variety of antibiotics. They further sequenced N1606 using a combination of MiSeq and Nanopore reads. The manuscript is well-written and in its good shape.
- Many thanks for your positive comments.
Major comments:
- This work sequenced the genome of an E. coli strain. Please briefly describe the library preparation process.
- It was added now as suggested by the reviewer.
- I would suggest using Galaxy (https://usegalaxy.org/) to assemble E. coli genomes. Galaxy offers free access. The analysis steps can be easily shared, and the analysis process is reproducible, the latter of which, in my opinion, matters significantly in this instance. The Galaxy communities published a preprint paper (https://doi.org/10.1101/347625) where an E. coli strain was fully sequenced and analyzed upon MiSeq and Nanopore reads.
- Many thanks for the reviewer comment.
- I would suggest adding a table showing the antibiotics tested and the MIC numbers.
- Many thanks for the reviewer comment. The table was added as suggested.
- Please further discuss why N1606 has accumulated so many antibiotics-resistant genes.
- The emergence of such XDR phenotype in E. coli strain due to accumulation of many plasmid associated multidrug-resistance determinants can be explained by the bacterial potential to acquire additional resistance traits through mobile genetic elements such as plasmids by horizontal gene transfer, not only among E. colistrains but also to other Enterobacterales. It is now discussed in the text.
Minor comments:
- Page 2 line 54, "from human urine sample" → "from the human urine sample"
- Revised as suggested.
- Page 2 line 71, "quality control" → "control".
- Revised as suggested.
- Page 2 line 82, "2634 bp to 120731 bp" → "2,634 bp to 120,731 bp".
- Revised as suggested.
Round 2
Reviewer 1 Report
If the authors do not have epidemiological data, this should be indicated as a limitation, right after the indication that this clone has already been detected in the environment of the same country.
The last conclusion they have added does not apply in this paper, as the authors do not provide any data on how the rapid tests have played any role in the clinical management of the patient. It should be deleted.
Author Response
Comments and Suggestions for Authors
If the authors do not have epidemiological data, this should be indicated as a limitation, right after the indication that this clone has already been detected in the environment of the same country.
R. Added as suggested (L147-148).
The last conclusion they have added does not apply in this paper, as the authors do not provide any data on how the rapid tests have played any role in the clinical management of the patient. It should be deleted.
R. It has been deleted as suggested by the reviewer.